# Bronchopulmonary Dysplasia in Extremely Premature Infants: A Scoping Review for Identifying Risk Factors

**DOI:** 10.3390/biomedicines11020553

**Published:** 2023-02-14

**Authors:** Masato Ito, Shin Kato, Makoto Saito, Naoyuki Miyahara, Hirokazu Arai, Fumihiko Namba, Erika Ota, Hidehiko Nakanishi

**Affiliations:** 1Department of Pediatrics, Akita University Graduate School of Medicine, Akita 010-8543, Japan; 2Department of Pediatrics and Neonatology, Nagoya City University Graduate School of Medical Sciences, Nagoya 467-8602, Japan; 3Department of Child Health, Faculty of Medicine, University of Tsukuba, Tsukuba 305-8546, Japan; 4Department of Pediatrics, Saitama Medical Center, Saitama Medical University, Kawagoe 350-8550, Japan; 5Department of Neonatology, Akita Red Cross Hospital, Akita 010-1495, Japan; 6Global Health Nursing, Graduate School of Nursing Sciences, St. Luke’s International University, Chuo 104-0044, Japan; 7Tokyo Foundation for Policy Research, Tokyo 106-6234, Japan; 8Research and Development Center for New Medical Frontiers, Department of Advanced Medicine, Division of Neonatal Intensive Care Medicine, Kitasato University School of Medicine, Sagamihara 252-0375, Japan

**Keywords:** bronchopulmonary dysplasia, scoping review, premature, neonates

## Abstract

Background: Over the years, bronchopulmonary dysplasia (BPD) affects the pulmonary function of infants, resulting in chronic health burdens for infants and their families. The aim of this scoping review was to screen available evidence regarding perinatal risk factors associated with the development and severity of BPD. Methods: The eligibility criteria of the studies were year of publication between 2016 and 2021; setting of a developed country; English or Japanese as the study language; and randomized controlled, cohort, or case-control design. The titles and abstracts of the studies were screened by independent reviewers. Results: Of 8189 eligible studies, 3 were included for severe BPD and 26 were included for moderate BPD. The risk factors for severe BPD were male sex, iatrogenic preterm birth, maternal hypertensive disorders of pregnancy (HDP), low gestational age, small-for-gestational-age (SGA) birth weight, mechanical ventilation on day 1, and need for patent ductus arteriosus (PDA) management. The risk factors for moderate or severe BPD included male sex, premature rupture of membranes, clinical chorioamnionitis, maternal HDP, SGA birth weight, bubbly/cystic appearance on X-ray, and PDA management. Conclusions: We identified several risk factors for BPD. We plan to confirm the validity of the new classification using the existing dataset.

## 1. Introduction

Bronchopulmonary dysplasia (BPD) is one of the most important chronic morbidities associated with prematurity [1,2]. It requires prolonged respiratory support and causes airway hypersensitivity [3] and obstructive pulmonary disease [4], resulting in an increased incidence of re-hospitalization [5], neurodevelopmental impairment [6], and pulmonary hypertension [7]. Over the years, these events have resulted in considerable chronic health burdens for infants and their families [8]. Due to significant improvements in the long-term prognosis of extremely premature infants [9], the number of BPD cases has increased. To avoid health deterioration and serious sequelae, it is important to identify the risk factors of BPD in premature infants and explore better prophylactic and treatment strategies for BPD.

Since BPD affects pulmonary function and neurodevelopmental outcomes, a classification that can be used to predict long-term prognosis is required. The BPD classification proposed in 2001 by the National Institute of Child Health and Human Development (NICHD) [10], which is based on clinical treatments, such as oxygen and ventilatory support, is widely used. However, the association between severity based on the NICHD classification and the long-term prognosis of infants is unclear, partly because the NICHD classification is consensus-based [11]. In 1992, a BPD classification based on etiology and chest X-ray findings was introduced in Japan [12]. An association between the Japanese classification and long-term outcomes has been reported [13], but to date, it is popular only in Japan [14]. We expect that unknown factors will fill the gaps in BPD definitions by combining the NICHD classification and the Japanese pathophysiology-based classification. Furthermore, the phenotype of BPD has changed since the introduction of antenatal corticosteroid and surfactant replacement therapy [15], and more premature newborns are managed using non-invasive positive pressure ventilation [16,17]. New modalities of respiratory support, such as the use of high-flow nasal cannulas, have also become popular [18]. Therefore, we believe that the Japanese pathophysiology-based BPD classification also needs to be revised to better reflect changes in phenotype and management [19].

We hope to develop a new pathophysiology-based BPD classification based on our original classification, which is internationally acceptable. The aim of this scoping review was to screen available evidence and identify perinatal risk factors associated with the development and severity of BPD to establish a novel BPD classification that can be used to predict long-term prognosis.

## 2. Materials and Methods

### 2.1. Search and Selection of Studies

With the aid of knowledgeable librarians, we created a search plan to locate eligible studies in PubMed and the Japanese database Ichushi Web. A search was conducted in October 2021 with no restrictions on date, time, language, document type, or publication status. Keywords were identified from experts’ opinions, literature review, controlled vocabulary (Cumulative Index to Nursing and Allied Health Literature Headings, Medical Subject Headings, Excerpta Medica Tree), and a review of primary search results. To obtain all search outcome results and avoid poor reporting of outcomes, we did not limit the search in any way. The search strategy was developed with the assistance of a medical information specialist.

### 2.2. Eligibility Criteria

Articles that met the following eligibility criteria were included: (1) publication between January 2002 and August 2021; (2) setting of a developed country; (3) English or Japanese as the study language; (4) randomized controlled, prospective/retrospective cohort, or case-control study design; (5) extremely premature infants born before 28 weeks of gestation as study participants; and (6) endpoint of severe BPD according to the NICHD classification as an indicator of poor respiratory prognosis. Since we considered the development in the pathological evaluation and intervention/treatment methods of BPD and the change in the risk factors of BPD over the last 20 years, we only included articles published between January 2016 and August 2021. Regarding the secondary objective, we included studies that involved participants with birth weight < 1500 g or gestational age (GA) < 32 weeks and with endpoint of moderate or severe BPD according to the NICHD classification or combined results of BPD or death. The following types of articles were excluded from the review: (1) letters, editorials, commentaries, unpublished manuscripts, dissertations, government reports, books and book chapters, conference proceedings, meeting abstracts, lectures and addresses, and consensus development statements; (2) articles with <500 registered newborns; (3) articles reported by countries with United Nations Human Development Index < 0.8; and (4) animal studies.

### 2.3. Selection of Sources of Evidence

We used the web application Rayyan (http://rayyan.qcri.org; accessed on 1 January 2022) to speed up the selection process of studies for inclusion in this scoping review. The search results were de-duplicated using EndNote 20 and sent to four researchers for screening and confirmation. Four authors (S.K., M.I., M.S., and N.M.) independently screened all the titles and abstracts to exclude non-eligible trials. Disagreements between the four authors were discussed and resolved with the assistance of another reviewer. We then conducted a full-text review of potentially relevant articles. Disagreements between the reviewers were discussed, and decisions regarding selection were made by consensus. The reference lists of the included articles were screened for primary studies that were missed by the search strategy. Following a full-text review, data (including data on the risk factors of BPD) were extracted from all selected studies. We attempted to contact the author (HA) to conduct an additional study and provide the data [20].

### 2.4. Data Charting

At least two reviewers independently charted the data of the included studies using an Excel data extraction form developed for this study, and they discussed the results. The following data were extracted: author information, title, journal, year of publication, identified risk factors, study design, and participants. The review protocol was published [21] and registered on the University Hospital Medical Information Network Clinical Trial Registry (registration number: UMIN000045529). This scoping review was reported in accordance with the Preferred Reporting Items for Systematic Reviews and Meta-Analyses Scoping Review Guideline [22].

## 3. Results

### 3.1. Risk Factors of Severe BPD

Of the 8189 eligible studies, three were included for risk factors of severe BPD (Figure 1).

The investigated prenatal risk factors of severe BPD included male sex [23], iatrogenic preterm birth (result of induced labor or a cesarean section before labor and/or rupture of membranes) [24], maternal hypertensive disorders of pregnancy (HDP) [23], low GA [23], and small-for-gestational-age (SGA) birth weight [23] (Table 1). Male sex (adjusted odds ratio [aOR]: 1.75, 95% confidence interval [CI]: 1.22–2.49), iatrogenic preterm birth (aOR: 1.90, 95% CI: 1.10–3.26), maternal HDP (aOR: 2.18; 95% CI: 1.45–3.28), low GA (aOR: 1.36, 95% CI: 1.19–1.55), and SGA birth weight (aOR: 3.25, 95% CI: 1.91–5.54) were found to be associated with a high risk of BPD. 

The investigated postnatal risk factors of severe BPD included mechanical ventilation on the first day after birth and the need for patent ductus arteriosus (PDA) management [23,25] (Table 2). Mechanical ventilation on the first day after birth was found to be associated with the onset of BPD (aOR: 2.84, 95% CI: 1.54–5.24). Furthermore, symptomatic PDA had a negative effect on BPD onset (aOR: 2.53, 95% CI: 1.41–4.53), but prophylactic treatment of PDA failed to show efficacy in BPD prophylaxis (aOR: 0.98, 95% CI: 0.53–1.81). 

With our original search settings, the number of studies that investigated the association between severe BPD and various factors was small. Therefore, we revisited our original settings and adjusted the conditions (i.e., birth weight < 1500 g or GA < 32 weeks), any BPD severity, or combined results of BPD or death.

### 3.2. Risk Factors for Moderate or Severe BPD

When the target infants had birth weight < 1500 g or GA < 32 weeks and the outcome of moderate or severe BPD, the investigated prenatal risk factors of BPD included infant sex [23,26,27,28], twins [20,26,29], delivery method [24,26,29], maternal body mass index [30], oligohydramnios [29], maternal docosahexaenoic acid (DHA) supplementation [31], antenatal corticosteroid therapy [26,29,32], premature rupture of membranes (PROM) [26,29], pathological chorioamnionitis [20,27], clinical chorioamnionitis [26,33], HDP [20,23,27,34], antepartum hemorrhage [35], GA [20,23,26,27,29,33], infant birth weight [20,26,29,33], SGA birth weight [20,23,27,33,36,37,38] and large-for-gestational-age birth weight [39] (Table 3). Male infants had a higher risk of BPD than female infants (four studies reported a positive association, and four studies evaluated the risk factors; i.e., 4/4, aOR: 1.75, 95% CI: 1.22–2.49 [23], aOR: 1.42, 95% CI: 1.32–1.53 [26], aOR: 1.39, 95% CI: 1.11–1.75 [27], aOR: 1.28, 95% CI: 1.06–1.54 [28]). PROM (2/2, aOR: 1.49, 95% CI: 1.04–2.14 [29], aOR: 1.11, 95% CI: 1.02–1.20 [26]), and pathological/clinical chorioamnionitis (4/4, aOR: 1.53, 95% CI: 1.21–1.94 [27]], aOR: 1.16, 95% CI: 1.04–1.30 [20], aOR: 1.34, 95% CI: 1.23–1.45 [33], aOR: 1.25, 95% CI: 1.14–1.37 [26]) tended to increase the risk of BPD. The association between SGA birth weight and BPD has been extensively studied, and it was reported in several studies that SGA birth weight is a risk factor for BPD (10/10, aOR: 5.65, 95% CI: 2.42–13.19 [27], aOR: 3.56, 95% CI: 3.04–4.18 [36], aOR: 3.35, 95% CI: 2.65–4.22 [37], aOR: 3.25, 95% CI: 1.91–5.54 [23], aOR: 2.82, 95% CI: 2.29–3.49 [36], aOR: 2.77, 95% CI: 2.23–3.43 [37], aOR: 1.84, 95% CI: 1.30–2.60 [36], aOR: 1.73, 95% CI: 1.56–1.91 [33], aOR: 1.30, 95% CI: 1.08.–1.54 [20], relative risk: 1.30, 95% CI: 1.20–1.40 [38]). 

The investigated postnatal risk factors of moderate or severe BPD included delayed cord clamping [40,41], resuscitation [33,42], respiratory management [23,33], bubbly/cystic appearance on X-ray [20], respiratory distress syndrome (RDS)/surfactant administration [20,23,27,33,43,44], caffeine administration [45], sivelestat administration [46], PDA management [23,25,27,33,47], infections [33], respiratory flora [48], breast milk [49], and infant DHA supplementation [50] (Table 4). Bubbly/cystic appearance on X-rays has been shown to have an adverse effect on BPD onset (aOR: 2.49, 95% CI: 2.24–2.77) [20]. Regarding RDS, two of the four extracted articles reported it as a risk factor of BPD (2/4, aOR: 2.44, 95% CI: 1.68–3.54 [23], aOR: 1.62, 95% CI: 0.96–2.72 [27], aOR: 1.24, 95% CI: 1.13–1.37 [33], aOR: 0.84, 95% CI: 0.74–0.94 [20]), but it was also reported that less invasive surfactant administration suppressed BPD onset (aOR: 0.55, 95% CI: 0.49–0.62) [44]. Most studies on PDA have reported that PDA has an adverse effect on BPD onset (3/4, aOR: 2.53, 95% CI: 1.41–4.53 [23] aOR: 2.30, 95% CI: 1.82–2.90 [27], aOR: 1.30, 95% CI: 1.20–1.41 [33], aOR: 0.47, 95% CI: 0.28–1.80 [47]). 

Table 5 and Table 6 show the extracted risk factors when the outcomes were combined results of BPD or death [51,52,53]. Male sex (1/1), low GA (1/1), SGA birth weight (1/1), and RDS (1/1) were selected as risk factors for moderate or severe BPD or death.

## 4. Discussion

To develop a new pathophysiology-based BPD classification that can be used to predict severe BPD early in life to facilitate early treatment, we conducted a scoping review and comprehensively evaluated the risk factors of BPD. We found that three previous studies were comparable to this study. The prenatal risk factors were male sex, iatrogenic preterm birth, maternal HDP, low GA, and SGA birth weight, and the postnatal risk factors were mechanical ventilation on the first day after birth and the need for PDA management. The risk factors of moderate or severe BPD included prenatal factors, such as male sex and SGA birth weight, and postnatal factors, such as RDS and PDA treatment. In addition, three of the included studies were conducted in Japan, and the risk factors mentioned in the studies included chorioamnionitis and bubble/cystic appearance. As previously reported, there are multiple risk factors for BPD; therefore, we reckoned that multiple factors should be extracted in this scoping review.

Unlike the international BPD classification, which is based on treatment, the Japanese BPD classification is based on etiological factors (e.g., RDS, intrauterine infection) and bubbly/cystic appearance on chest X-ray [12]. Imaging practices in neonatal intensive care units have not evolved significantly since Northway et al. reported an overview of BPD. Hirata et al. reported that bubbly/cystic appearance on chest X-rays was associated with respiratory prognosis [13]. We also found that bubbly/cystic appearance on chest X-ray was a risk factor for moderate or severe BPD. Therefore, we consider bubbly/cystic appearance on chest X-ray as a factor that can be used to develop a new BPD classification that can be used to predict severe BPD early in life, thereby facilitating early treatment.

Male sex had been reported to be a risk factor for BPD even before the articles included in this scoping review were published [54,55,56,57], and the results of our review are comparable to those of previous studies. Many articles on the relationship between SGA birth weight and BPD were extracted. It has been reported that biological mechanisms such as placental dysfunction and deficiency of insulin growth factor, vascular endothelial growth factor (VEGF), and VEGF receptors, which result in intrauterine growth retardation, can lead to fetal lung growth retardation [58].

In this scoping review, there were many studies that reported RDS as a risk factor for moderate or severe BPD, but some studies had contrary reports. In Japan, many institutions are attempting to improve the diagnosis of RDS by incorporating the prediction of surfactant secretion using a stable microbubble test [59,60]. However, the latest European guidelines recommend that surfactants be administered for RDS if sustained positive pressure ventilation fails [61]. This recommendation takes into account physical and radiographic findings but not surfactant secretion. Moreover, considering that most extremely premature infants are administered surfactants after birth using different methods [62,63,64], RDS may no longer be a necessary factor in the future diagnosis of BPD in extremely premature infants.

There is yet no consensus on whether chorioamnionitis increases the odds of developing BPD [14,65,66,67]. PROM is regarded as the leakage of amniotic fluid before the onset of labor, and chorioamnionitis is an important factor [68]. It was suggested that the importance of chorioamnionitis in the diagnosis of BPD differs from country to country, and in this study, we found reviews of studies conducted in Japan and South Korea. Data on chorioamnionitis are extracted when a database is built in these two above-mentioned countries, which is an advantage. Pathological chorioamnionitis is thought to be associated with a high incidence of PROM and preterm birth [69]. However, symptoms of clinical chorioamnionitis and pathological chorioamnionitis are not always simultaneously present [70]. It was reported that the positive predictive value of pathological chorioamnionitis was high for uterine fundal tenderness and purulent or foul amniotic fluid or cervical discharge, but significantly low for maternal fever [71]. If chorioamnionitis is to be considered an onset factor of BPD, it may be necessary to decide on a more rigorous diagnostic method and selection criteria.

Due to factors such as pulmonary edema and endothelial injury because of increased pulmonary blood flow, increased risk of pulmonary hemorrhage, and the necessity of prolonged and aggressive ventilation, PDA is thought to have an adverse effect on the development of BPD [72]. Regarding PDA, there are various views on the diagnostic criteria, treatment methods, and indications for ligation. There are also reports that ibuprofen administration did not reduce the risk of developing BPD [73], or even increased it [74]. It is necessary to continue to focus on BPD prevention.

A limitation of this scoping review is the differences in the study design and patient characteristics between the articles. For practical reasons, this scoping review considered only articles written in English and Japanese. We also searched for articles in Japanese, but there were no such articles that were subject to review. Furthermore, due to the nature of the outlined search methodology, important published studies may have been omitted.

In conclusion, in this scoping review, we identified various prenatal and postnatal risk factors for moderate or severe BPD. Studies conducted in Japan and South Korea showed that chest X-ray findings and chorioamnionitis were predictors of BPD severity. These two factors should be combined to construct a new BPD classification that can be used to predict prognosis. In the future, we plan to use registry data to verify the prognosis predicted using the new BPD classification.

## Figures and Tables

**Figure 1 biomedicines-11-00553-f001:**
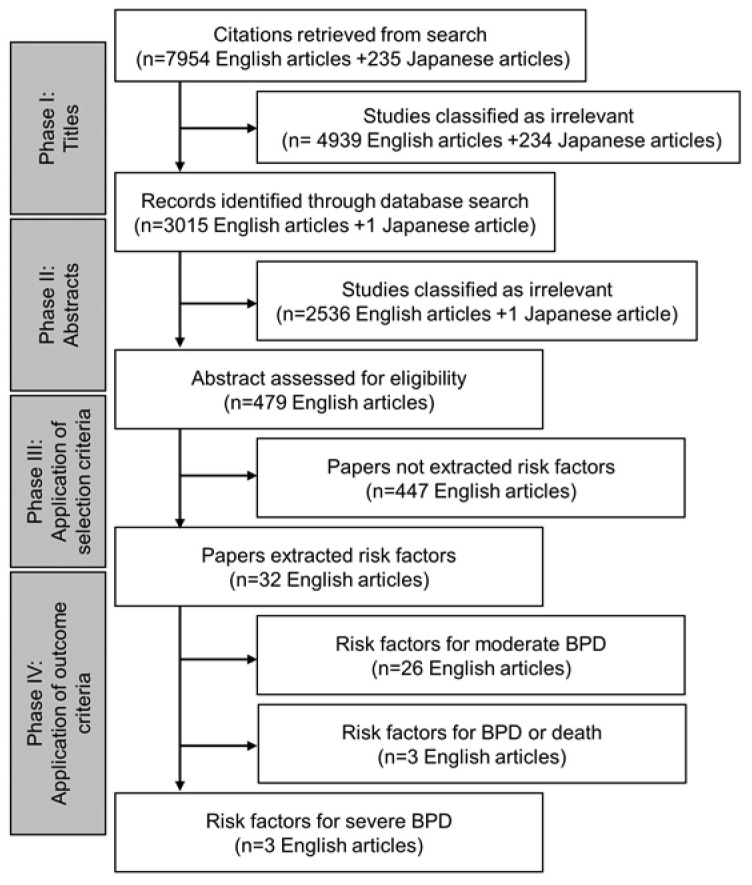
Study flow chart.

**Table 1 biomedicines-11-00553-t001:** Prenatal risk factors of severe BPD.

Risk Factor	Number of Studies with Positive Association/Number of Studies That Evaluated Risk Factors	aOR or RR*	95% CI	Author, Year	Country	Study Design	Number of Participants	Patient Characteristic	BPD Definition	Reference Number
Male sex	1/1	1.75	1.22–2.49	Tagliaferro, T., 2019	USA	Retrospective cohort study	1218	Birth at 23–28 wks GA	need for ≥30% O2 at 36 wks PMA	[23]
Iatrogenic preterm birth	1/1	1.90	1.10–3.26	Fritz, T., 2018	Sweden	Prospective cohort study	737	Birth at <27 wks GA	need for ≥30% O2 at 36 wks PMA	[24]
HDP	1/1	2.18	1.45–3.28	Tagliaferro, T., 2019	USA	Retrospective cohort study	1218	Birth at 23–28 wks GA	need for ≥30% O2 at 36 wks PMA	[23]
Low GA	1/1	1.36	1.19–1.55	Tagliaferro, T., 2019	USA	Retrospective cohort study	1218	Birth at 23–28 wks GA	need for ≥30% O2 at 36 wks PMA	[23]
SGA birth weight	1/1	3.25	1.91–5.54	Tagliaferro, T., 2019	USA	Retrospective cohort study	1218	Birth at 23–28 wks GA	need for ≥30% O2 at 36 wks PMA	[23]

BPD: bronchopulmonary dysplasia, aOR: adjusted odds ratio, RR: relative risk, CI: confidence interval, HDP: hypertensive disorders of pregnancy, GA: gestational age, SGA: small for gestational age, wks: weeks, PMA: postmenstrual age.

**Table 2 biomedicines-11-00553-t002:** Postnatal risk factors of severe BPD.

Risk Factor	Number of Studies with Positive Association/Number of Studies That Evaluated Risk Factors	aOR or RR*	95% CI	Author, Year	Country	Study Design	Number of Participants	Patient Characteristic	BPD Definition	Reference Number
Mechanical ventilation on first day after birth	1/1	2.84	1.54–5.24	Tagliaferro, T., 2019	USA	Retrospective cohort study	1218	Birth at 23–28 wks GA	need for ≥30% O2 at 36 wks PMA	[23]
Prophylactic treatment for PDA	0/1	0.98	0.53–1.81	Shin, J., 2021	Korea	Prospective cohort study	2303	Birth at <28 wks GA	need for ≥30% O2 at 36 wks PMA	[25]
Need for PDA management	1/1	2.53	1.41–4.53	Tagliaferro, T., 2019	USA	Retrospective cohort study	1218	Birth at 23–28 wks GA	need for ≥30% O2 at 36 wks PMA	[23]

BPD: bronchopulmonary dysplasia, aOR: adjusted odds ratio, RR: relative risk, CI: confidence interval, PDA: patent ductus arteriosus, wks: weeks, GA: gestational age, wks: weeks, PMA: postmenstrual age.

**Table 3 biomedicines-11-00553-t003:** Prenatal risk factors of moderate or severe BPD.

Risk Factor	Number of Studies with positive Association /Number of Studies that evaluated Risk Factors	aOR or RR*	95% CI	Author, Year	Country	Study Design	Number of Participants	Patient Characteristic	BPD Definition	Reference Number
Male sex	4/4	1.75	1.22–2.49	Tagliaferro, T., 2019	USA	Retrospective cohort study	1218	Birth at 23–28 wks GA	need for ≥30% O2 at 36 wks PMA	[23]
		1.42	1.32–1.53	Ushida, T., 2021	Japan	Retrospective cohort study	31,157	Birth at 22–31 wks GA with VLBW	need for >21% O2 at 36 wks PMA	[26]
		1.39	1.11–1.75	Shin, S. H., 2020	Korea	Prospective cohort study	2276	Birth at <30 wks GA	need for >21% O2 or positive pressure support at 36 wks PMA	[27]
		1.28	1.06–1.54	Shim, S. Y., 2017	Korea	Retrospective cohort study	1839	Birth at 23–29 wks GA with VLBW	need for >21% O2	[28]
Monochorionic twins	1/1	1.34	1.18–1.52	Ushida, T., 2021	Japan	Retrospective cohort study	31,157	Birth at 22–31 wks GA with VLBW	need for >21% O2 at 36 wks PMA	[26]
Multiple birth	0/2	0.82	0.71–0.94	Arai, H., 2019	Japan	Retrospective cohort study	15,480	BPD, birth at <28 wks GA with BW < 1500 g	need for >21% O2 or positive pressure support at 36 wks PMA	[20] **
		0.62	0.40–0.96	Park, J. H., 2021	Korea	Retrospective cohort study	884	Birth at 23–27 wks GA with PROM	need for >21% O2 or positive pressure support at 36 wks PMA	[29]
Iatrogenic preterm birth	1/1	1.90	1.10–3.26	Fritz, T., 2018	Sweden	Prospective cohort study	737	Birth at <27 wks GA	need for ≥30% O2 at 36 wks PMA	[24]
Vaginal delivery	0/1	0.85	0.78–0.94	Ushida, T., 2021	Japan	Retrospective cohort study	31,157	Birth at 22–31 wks GA with VLBW	need for >21% O2 at 36 wks PMA	[26]
CS	0/1	1.39	0.95–2.03	Park, J. H., 2021	Korea	Retrospective cohort study	884	Birth at 23–27 wks GA with PROM	need for >21% O2 or positive pressure support at 36 wks PMA	[29]
High maternal BMI	0/1	1.22	0.96–1.56	Lee, B. K., 2021	Korea	Retrospective cohort study	7348	Birth at 25–30 wks GA with VLBW	need for ≥30% O2 or positive pressure support at 36 wks PMA	[30]
Low maternal BMI	1/1	1.44	1.06–1.96	Lee, B. K., 2021	Korea	Retrospective cohort study	7348	Birth at 25–30 wks GA with VLBW	need for ≥30% O2 or positive pressure support at 36 wks PMA	[30]
Oligohydramnios	0/1	1.13	0.76–1.69	Park, J. H., 2021	Korea	Retrospective cohort study	884	Birth at 23–27 wks GA with PROM	need for >21% O2 or positive pressure support at 36 wks PMA	[29]
Maternal docosahexaenoic acid supplementation	0/1	0.91 *	0.80–1.04	Marc, I., 2020	Canada	RCT	528	Birth at 29 wks GA	need for >21% O2 at 36 wks PMA	[31]
Antenatal corticosteroid therapy	0/2	0.96	0.90–1.03	Travers, C. P., 2018	USA	Retrospective cohort study	9715	Birth at 22–28 wks GA with BW > 400 g	need for >21% O2 at 36 wks PMA	[32]
		0.78	0.41–1.49	Park, J. H., 2021	Korea	Retrospective cohort study	884	Birth at 23–27 wks GA with PROM	need for >21% O2 or positive pressure support at 36 wks PMA	[29]
No antenatal corticosteroid therapy	0/1	0.72	0.67–0.78	Ushida, T., 2021	Japan	Retrospective cohort study	31,157	Birth at 22–31 wks GA with VLBW	need for >21% O2 at 36 wks PMA	[26]
PROM	2/2	1.49	1.04–2.14	Park, J. H., 2021	Korea	Retrospective cohort study	884	Birth at 23–27 wks GA with PROM	need for >21% O2 or positive pressure support at 36 wks PMA	[29]
		1.11	1.02–1.20	Ushida, T., 2021	Japan	Retrospective cohort study	31,157	Birth at 22–31 wks GA with VLBW	need for >21% O2 at 36 wks PMA	[26]
Pathological chorioamnionitis	2/2	1.53	1.21–1.94	Shin, S. H., 2020	Korea	Prospective cohort study	2276	Birth at <30 wks GA	need for >21% O2 or positive pressure support at 36 wks PMA	[27]
		1.16	1.04–1.30	Arai, H., 2019	Japan	Retrospective cohort study	15,480	BPD, birth at <28 wks GA with BW < 1500 g	need for >21% O2 or positive pressure support at 36 wks PMA	[20] **
Clinical chorioamnionitis	2/2	1.34	1.23–1.45	Nakashima, T., 2020	Japan	Retrospective cohort study	17,126	Birth at 22–27 wks GA	need for >21% O2 or positive pressure support at 36 wks PMA	[33]
		1.25	1.14–1.37	Ushida, T., 2021	Japan	Retrospective cohort study	31,157	Birth at 22–31 wks GA with VLBW	need for >21% O2 at 36 wks PMA	[26]
HDP	3/4	2.18	1.45–3.28	Tagliaferro, T., 2019	USA	Retrospective cohort study	1218	Birth at 23–28 wks GA	need for ≥30% O2 at 36 wks PMA	[23]
		1.47	1.03–2.12	Shin, S. H., 2020	Korea	Prospective cohort study	2276	Birth at <30 wks GA	need for >21% O2 or positive pressure support at 36 wks PMA	[27]
		1.33	1.09–1.61	Arai, H., 2019	Japan	Retrospective cohort study	15,480	BPD, birth at <28 wks GA with BW < 1500 g	need for >21% O2 or positive pressure support at 36 wks PMA	[20] **
		0.75	0.58–0.97	Sloane, A. J., 2019	USA	Retrospective cohort study	5456	VLBW	need for >21% O2 at 36 wks PMA	[34]
Antepartum hemorrhage	0/1	1.19	0.98–1.43	Klinger, G., 2021	Israel	Retrospective cohort study	33,627	Birth at 24–31 wks GA with VLBW	need for >21% O2 at 36 wks PMA	[35]
Low GA	1/1	1.36	1.19–1.55	Tagliaferro, T., 2019	USA	Retrospective cohort study	1218	Birth at 23–28 wks GA	need for ≥30% O2 at 36 wks PMA	[23]
High GA	0/3	0.97	0.96–0.97	Ushida, T., 2021	Japan	Retrospective cohort study	31,157	Birth at 22–31 wks GA with VLBW	need for >21% O2 at 36 wks PMA	[26]
		0.78	0.62–0.99	Park, J. H., 2021	Korea	Retrospective cohort study	884	Birth at 23–27 wks GA with PROM	need for >21% O2 or positive pressure support at 36 wks PMA	[29]
		0.67	0.63–0.72	Shin, S. H., 2020	Korea	Prospective cohort study	2276	Birth at <30 wks GA	need for >21% O2 or positive pressure support at 36 wks PMA	[27]
GA < 26 wks	2/2	2.68	2.47–2.91	Nakashima, T., 2020	Japan	Retrospective cohort study	17,126	Birth at 22–27 wks GA	need for >21% O2 or positive pressure support at 36 wks PMA	[33]
		1.30	1.07–1.47	Arai, H., 2019	Japan	Retrospective cohort study	15,480	BPD, birth at <28 wks GA with BW < 1500 g	need for >21% O2 or positive pressure support at 36 wks PMA	[20] **
High BW	0/2	1.00	1.00–1.00	Park, J. H., 2021	Korea	Retrospective cohort study	884	Birth at 23–27 wks GA with PROM	need for >21% O2 or positive pressure support at 36 wks PMA	[29]
		0.77	0.76–0.79	Ushida, T., 2021	Japan	Retrospective cohort study	31,157	Birth at 22–31 wks GA with VLBW	need for >21% O2 at 36 wks PMA	[26]
BW < 700 g	1/1	1.71	1.43–2.06	Arai, H., 2019	Japan	Retrospective cohort study	15,480	BPD, birth at <28 wks GA with BW < 1500 g	need for >21% O2 or positive pressure support at 36 wks PMA	[20] **
BW < 750 g	1/1	3.14	2.90–3.41	Nakashima, T., 2020	Japan	Retrospective cohort study	17,126	Birth at 22–27 wks GA	need for >21% O2 or positive pressure support at 36 wks PMA	[33]
SGA BW	10/10	5.65	2.42–13.19	Shin, S. H., 2020	Korea	Prospective cohort study	2276	Birth at <30 wks GA	need for >21% O2 or positive pressure support at 36 wks PMA	[27]
		3.56	3.04–4.18	Boghossian, N. S., 2018	USA	Retrospective cohort study	data not available	Birth at 27 wks GA	need for >21% O2 at 36 wks PMA	[36]
		3.35	2.65–4.22	Aldana-Aguirre, J. C., 2019	Canada	Retrospective cohort study	5309	PDA, birth at <33 wks GA	need for >21% O2 at 36 wks PMA	[37]
		3.25	1.91–5.54	Tagliaferro, T., 2019	USA	Retrospective cohort study	1218	Birth at 23–28 wks GA	need for ≥30% O2 at 36 wks PMA	[23]
		2.82	2.29–3.49	Boghossian, N. S., 2018	USA	Retrospective cohort study	data not available	Birth at 29 wks GA	need for >21% O2 at 36 wks PMA	[36]
		2.77	2.23–3.43	Aldana-Aguirre, J. C., 2019	Canada	Retrospective cohort study	16,998	Birth at <33 wks GA	need for >21% O2 at 36 wks PMA	[37]
		1.84	1.30–2.60	Boghossian, N. S., 2018	USA	Retrospective cohort study	data not available	Birth at 23 wks GA	need for >21% O2 at 36 wks PMA	[36]
		1.73	1.56–1.91	Nakashima, T., 2020	Japan	Retrospective cohort study	17,126	Birth at 22–27 wks GA	need for >21% O2 or positive pressure support at 36 wks PMA	[33]
		1.29	1.08–1.54	Arai, H., 2019	Japan	Retrospective cohort study	15,480	BPD, birth at <28 wks GA with BW < 1500 g	need for >21% O2 or positive pressure support at 36 wks PMA	[20] **
		1.30 *	1.20–1.40	Monier, I., 2017	France	Retrospective cohort study	2505	Birth at <32 wks GA	need for >21% O2 or positive pressure support at 36 wks PMA	[38]
LGA BW	0/1	0.71	0.68–0.73	Boghossian, N. S., 2018	USA	Retrospective cohort study	156,587	Birth at 22–29 wks GA	need for >21% O2 at 36 wks PMA	[39]

** reanalyzed and extracted as risk factors. BPD: bronchopulmonary dysplasia, aOR: adjusted odds ratio, RR: relative risk, CI: confidence interval, CS: cesarean section, BMI: body mass index, PROM: premature rupture of membranes, pPROM: preterm premature rupture of membranes, HDP: hypertensive disorders of pregnancy, wks: weeks, GA: gestational age, BW: birth weight, VLBW: very low birth weight, SGA: small for gestational age, LGA: large for gestational age, PMA: postmenstrual age.

**Table 4 biomedicines-11-00553-t004:** Postnatal risk factors of moderate or severe BPD.

Risk Factor	Number of Studies with positive Association /Number of Studies That Evaluated Risk Factors	aOR or RR*	95% CI	Author, Year	Country	Study Design	Number of Participants	Patient Characteristic	BPD Definition	Reference Number
Delayed cord clamping	0/2	1.04 *	0.95–1.14	Tarnow-Mordi, W., 2017	Australia	Randomized clinical trial	1634	Birth at <32 wks GA	need for >21% O2 or positive pressure support at 36 wks PMA	[40]
		0.99	0.86–1.14	Lodha, A., 2019	Canada	Retrospective cohort study	4680	Birth at 22–28 wks GA	need for >21% O2 at 36 wks PMA	[41]
Extensive cardiopulmonary resuscitation	1/1	1.68	1.19–2.37	Shukla, V., 2020	Canada	Retrospective cohort study	3633	Birth at <26 wks GA	need for >21% O2 at 36 wks PMA or at discharge from the participating unit	[42]
Intubation at birth	1/1	1.84	1.64–2.07	Nakashima, T., 2020	Japan	Retrospective cohort study	17,126	Birth at 22–27 wks GA	need for >21% O2 or positive pressure support at 36 wks PMA	[33]
Mechanical ventilation on first day after birth	1/1	2.84	1.54–5.24	Tagliaferro, T., 2019	USA	Retrospective cohort study	1218	Birth at 23–28 wks GA	need for ≥30% O2 at 36 wks PMA	[23]
Supplemental oxygen for >4 weeks	1/1	6.98	8.27–11.72	Nakashima, T., 2020	Japan	Retrospective cohort study	17,126	Birth at 22–27 wks GA	need for >21% O2 or positive pressure support at 36 wks PMA	[33]
Invasive ventilation for >4 weeks	1/1	4.82	4.39–5.30	Nakashima, T., 2020	Japan	Retrospective cohort study	17,126	Birth at 22–27 wks GA	need for >21% O2 or positive pressure support at 36 wks PMA	[33]
Non-invasive positive pressure ventilation for >4 weeks	1/1	1.11	1.03–1.20	Nakashima, T., 2020	Japan	Retrospective cohort study	17,126	Birth at 22–27 wks GA	need for >21% O2 or positive pressure support at 36 wks PMA	[33]
Bubbly/cystic appearance on X-ray	1/1	2.49	2.24–2.77	Arai, H., 2019	Japan	Retrospective cohort study	15,480	BPD, birth at <28 wks GA with BW < 1500 g	need for >21% O2 or positive pressure support at 36 wks PMA	[20] **
RDS/surfactant administration	2/4	2.44	1.68–3.54	Tagliaferro, T., 2019	USA	Retrospective cohort study	1218	Birth at 23–28 wks GA	need for ≥30% O2 at 36 wks PMA	[23]
		1.62	0.96–2.72	Shin, S. H., 2020	Korea	Prospective cohort study	2276	Birth at <30 wks GA	need for >21% O2 or positive pressure support at 36 wks PMA	[27]
		1.24	1.13–1.37	Nakashima, T., 2020	Japan	Retrospective cohort study	17,126	Birth at 22–27 wks GA	need for >21% O2 or positive pressure support at 36 wks PMA	[33]
		0.84	0.74–0.94	Arai, H., 2019	Japan	Retrospective cohort study	15,480	BPD, birth at <28 wks GA with BW < 1500 g	need for >21% O2 or positive pressure support at 36 wks PMA	[20] **
Late surfactant administration	1/1	1.55	1.20–2.00	Stritzke, A., 2018	Canada	Retrospective cohort study	2512	Birth at <28 wks GA	need for >21% O2 at 36 wks PMA	[43]
Less invasive surfactant administration	0/1	0.55	0.49–0.62	Härtel, C., 2018	Germany	Retrospective cohort study	7533	Birth at 22–29 wks GA with BW < 1500 g	need for >21% O2 or positive pressure support at 36 wks PMA	[44]
Early caffeine administration	0/1	0.61	0.45–0.81	Lodha, A., 2019	Canada	Retrospective cohort study	3993	Birth at <29 wks GA	need for >21% O2 at 36 wks PMA	[45]
Sivelestat administration	0/1	0.83	0.53–1.30	Ogawa, R., 2017	Japan	Retrospective cohort study	1031	Birth at <28 wks GA with BW < 1000 g	need for >21% O2 at 28 days old or 36 wks PMA	[46]
Prophylactic treatment for PDA	0/1	0.98	0.53–1.81	Shin, J., 2021	Korea	Prospective cohort study	2303	Birth at <28 wks GA	need for ≥30% O2 at 36 wks PMA	[25]
Need for PDA management	3/4	2.53	1.41–4.53	Tagliaferro, T., 2019	USA	Retrospective cohort study	1218	Birth at 23–28 wks GA	need for ≥30% O2 at 36 wks PMA	[23]
		2.30	1.82–2.90	Shin, S. H., 2020	Korea	Prospective cohort study	2276	Birth at <30 wks GA	need for >21% O2 or positive pressure support at 36 wks PMA	[27]
		1.30	1.20–1.41	Nakashima, T., 2020	Japan	Retrospective cohort study	17,126	Birth at 22–27 wks GA	need for >21% O2 or positive pressure support at 36 wks PMA	[33]
		0.47	0.28–1.80	Mohamed, M. A., 2017	USA	Retrospective cohort study	643	BW < 1500 g	need for >21% O2 at 36 wks PMA	[47]
Sepsis	1/1	1.50	1.33–1.69	Nakashima, T., 2020	Japan	Retrospective cohort study	17,126	Birth at 22–27 wks GA	need for >21% O2 or positive pressure support at 36 wks PMA	[33]
Normal respiratory flora	0/1	0.58	0.34–0.99	Antoine, J., 2020	Australia	Retrospective cohort study	1054	Birth at <32 wks GA	need for >21% O2 or positive pressure support at 36 wks PMA	[48]
Breast milk	0/1	0.40	0.27–0.67	Dicky, O., 2017	France	Prospective cohort study	926	Birth at <32 wks GA	need for >21% O2 or positive pressure support at 36 wks PMA	[49]
Enteral docosahexaenoic acid supplementation	1/1	1.13	1.02–1.25	Collins, C. T., 2017	Australia	RCT	1273	Birth at <29 wks GA	need for >21% O2 or positive pressure support at 36 wks PMA	[50]

** Reanalyzed and extracted as risk factors. BPD: bronchopulmonary dysplasia, aOR: adjusted odds ratio, RR: relative risk, CI: confidence interval, RDS: respiratory distress syndrome, PDA: patent ductus arteriosus, RCT: randomized clinical trial, wks: weeks, GA: gestational age, BW: birth weight, PMA: postmenstrual age.

**Table 5 biomedicines-11-00553-t005:** Prenatal risk factors of moderate or severe BPD or death.

Risk Factor	Number of Studies with Positive Association/Number of Studies That Evaluated Risk Factors	aOR or RR*	95% CI	Author, Year	Country	Study Design	Number of Participants	Patient Characteristic	BPD Definition	Reference Number
Male sex	1/1	1.37	1.19–1.58	Jung, Y. H., 2019	Korea	Prospective cohort study	4940	Birth at 23–31 wks GA	need for >21% O2 at 36 wks PMA	[51]
Oligohydramnios	0/1	1.22	1.00–1.50	Jung, Y. H., 2019	Korea	Prospective cohort study	4940	Birth at 23–31 wks GA	need for >21% O2 at 36 wks PMA	[51]
LGA birth weight	1/1	1.70	1.63–1.77	Jung, Y. H., 2019	Korea	Prospective cohort study	4940	Birth at 23–31 wks GA	need for >21% O2 at 36 wks PMA	[51]
SGA z-score	1/1	1.35	1.20–1.52	Jung, Y. H., 2019	Korea	Prospective cohort study	4940	Birth at 23–31 wks GA	need for >21% O2 at 36 wks PMA	[51]
Short length at birth z-score	1/1	1.25	1.14–1.37	Jung, Y. H., 2019	Korea	Prospective cohort study	4940	Birth at 23–31 wks GA	need for >21% O2 at 36 wks PMA	[51]

BPD: bronchopulmonary dysplasia, aOR: adjusted odds ratio, RR: relative risk, CI: confidence interval, LGA: large for gestational age, SGA: small for gestational age, wks: weeks, GA: gestational age, PMA: postmenstrual age.

**Table 6 biomedicines-11-00553-t006:** Postnatal risk factors of moderate or severe BPD or death.

Risk Factor	Number of Studies with Positive Association/Number of Studies That Evaluated Risk Factors	aOR or RR*	95% CI	Author, Year	Country	Study Design	Number of Participants	Patient Characteristic	BPD Definition	Reference Number
Hypothemia	0/1	1.03	0.64–1.68	Jensen, C. F., 2017	Denmark	Retrospective cohort study	808	Birth at <32 wks GA	need for >21% O2 at 36 wks PMA	[52]
RDS	1/1	1.90	1.46–2.48	Jung, Y. H., 2019	Korea	Prospective cohort study	4940	Birth at 23–31 wks GA	need for >21% O2 at 36 wks PMA	[51]
≤48 hours of antibiotic therapy	0/1	1.19	0.88–1.62	Flannery, D. D., 2018	USA	Retrospective cohort study	4950	Sepsis, birth at <32 wks GA with BW <1500 g	need for >21% O2 or positive pressure support at 36 wks PMA	[53]
3–7 days of antibiotic therapy	0/1	0.82	0.65–1.04	Flannery, D. D., 2018	USA	Retrospective cohort study	4950	Sepsis, birth at <32 wks GA with BW <1500 g	need for >21% O2 or positive pressure support at 36 wks PMA	[53]

BPD: bronchopulmonary dysplasia, aOR: adjusted odds ratio, RR: relative risk, CI: confidence interval, RDS: respiratory distress syndrome, wks: weeks, GA: gestational age, BW: birth weight, PMA: postmenstrual age.

## Data Availability

Not applicable.

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
