# Peer review of "Bronchopulmonary Dysplasia in Extremely Premature Infants: A Scoping Review for Identifying Risk Factors"

_biomedicines, 2023, doi:10.3390/biomedicines11020553_

Round 1
Reviewer 1 Report
I have read this paper with great interest, and with a background of clinical research in neonatology.
Although I understand the intention (lines 60-62) to come up with another BPD definition, I cannot connect this with the analysis currently done (perinatal risk factors for BPD (severity) as the classification and search has been done based on the NICHD (cf methods). How can this subsequently result in another classification ? You repeat this in the first lines of the discussion, but I simply do not yet understand the rationale, or reasoning for this approach taken (how can a search for risk factors based on NICHD result in a new classification ?}
For reasons of clarity, I suggest to add the NICHD classification (cfr severe) to the text section on eligibility criteria, perhaps as a table ?
Editing comment: sorry, but figure 1 cannot be read properly, same reflections for the tables, and perhaps use subheading (prenatal and postnatal) in the results section.
Please add the definition of ‘iatrogenic preterm birth’
As there is new information on the association of pda treatment and bpd (cfr beneduct study, ibuprofen associated with a higher BPD risk), can you add information on the pharmacological treatments for PDA ?
Line 166, risk factors or protective (cord clamping)
In the methods section, you see to focus on NICHD classification, but in the discussion, there is a shift to the Japanese classification, and you mix both classification (cfr conclusion section). How can you do this if only one classification has been explored in the systematic review ?
Author Response
Answer to reviewer 1
The authors really appreciate the rigorous efforts and expertise of the reviewer in reviewing our manuscript.
â‘ Although I understand the intention (lines 60-62) to come up with another BPD definition, I cannot connect this with the analysis currently done (perinatal risk factors for BPD (severity) as the classification and search has been done based on the NICHD (cf methods). How can this subsequently result in another classification? You repeat this in the first lines of the discussion, but I simply do not yet understand the rationale, or reasoning for this approach taken (how can a search for risk factors based on NICHD result in a new classification ?}
Answerâ‘ : Thank you for this suggestion. As described in the introduction, Japanese original BPD classification is a pathophysiology-based classification, which is separate from the NICHD classification. To attempt a scoping review and make data extraction more effective, we used the NICHD criteria as an indicator of poor respiratory outcome.
We have revised the introduction, methods and discussion.
â‘¡For reasons of clarity, I suggest to add the NICHD classification (cfr severe) to the text section on eligibility criteria, perhaps as a table ?
Answerâ‘¡: Thank you for this suggestion. As mentioned above, the purpose of this scoping review was to revise the pathophysiological-based classification of BPD in the Japanese population. And the NICHD classification was used as an indicator of poor respiratory outcome in reviewing the literature. Therefore, the details of the NICHD classification are not described in the text.
â‘¢Please add the definition of ‘iatrogenic preterm birth’
Answerâ‘¢: Thank you for this suggestion. We have added the definition of iatrogenic preterm birth (lines 125-126).
As there is new information on the association of pda treatment and bpd (cfr beneduct study, ibuprofen associated with a higher BPD risk), can you add information on the pharmacological treatments for PDA?
Answerâ‘£: Thank you for this suggestion. We have added a discussion of the relationship between ibuprofen and BPD (lines 250-251)
⑤Line 166, risk factors or protective (cord clamping).
Answer⑤:Thank you for this suggestion. It is risk factor.
â‘¥In the methods section, you see to focus on NICHD classification, but in the discussion, there is a shift to the Japanese classification, and you mix both classification (cfr conclusion section). How can you do this if only one classification has been explored in the systematic review?
Answer⑥:Thank you for this suggestion. As mentioned above, the purpose of this scoping review was to revise the pathophysiological-based classification of BPD in the Japanese population. And the NICHD classification was used as an indicator of poor respiratory outcome in reviewing the literature.

Reviewer 2 Report
This is an interesting and generally well written manuscript. Only few points deserve to be improved. In particular:
Tables must be improved because are barely readable
Lines 233-239: authors must define PROM. Moreover, the link between PROM and chorioamnionitis deserves to be highlithed. In fact, the inflammatory cytokines found in the amniotic fluid during chorioamnionitis alter the cell-cell junctions weaking placental membrane favoring PROM (see PMID: 26739007).
Abbreviations must be written in full length when mentioned for the first time
Author Response
Answer to reviewer 2
The authors really appreciate the rigorous efforts and expertise of the reviewer in reviewing our manuscript.
- Tables must be improved because are barely readable
Answer①:Thank you for this suggestion. We plan to work with the editors to improve them.
- Lines 233-239: authors must define PROM. Moreover, the link between PROM and chorioamnionitis deserves to be highlithed. In fact, the inflammatory cytokines found in the amniotic fluid during chorioamnionitis alter the cell-cell junctions weaking placental membrane favoring PROM (see PMID: 26739007).
Answer②:Thank you for this suggestion. We have added the consideration (lines 233-234).
- Abbreviations must be written in full length when mentioned for the first time
Answer③:Thank you for this suggestion. We have checked again.

Round 2
Reviewer 1 Report
thank you for the revision, the rationale and hypothesis are clearer now in my assessment. I only have one minor specific suggestion: what do you mean with the conclusion section of both the abstract and the full paper ? do you mean that you will explore an existing dataset on the usefullness of the risk factors identified to 'predict' or who are 'associated' with BPD outcome ?
Author Response
Answer to reviewer
The authors really appreciate the rigorous efforts and expertise of the reviewer in reviewing our manuscript.
- thank you for the revision, the rationale and hypothesis are clearer now in my assessment. I only have one minor specific suggestion: what do you mean with the conclusion section of both the abstract and the full paper ? do you mean that you will explore an existing dataset on the usefullness of the risk factors identified to 'predict' or who are 'associated' with BPD outcome ?
Answerâ‘
We are creating a new pathophysiology-based BPD classification based on these results. We plan to confirm the validity of the new classification using the existing dataset. We have revised the abstract.
